# Screening for Resistance Resources against Bacterial Wilt in Wild Potato

**DOI:** 10.3390/plants13020220

**Published:** 2024-01-13

**Authors:** Wenfeng He, Bingsen Wang, Mengshu Huang, Chengzhen Meng, Jiahui Wu, Juan Du, Botao Song, Huilan Chen

**Affiliations:** 1National Key Laboratory for Germplasm Innovation & Utilization of Horticultural Crops, Wuhan 430070, China; hewenfeng0915@163.com (W.H.); wangbingsen@webmail.hzau.edu.cn (B.W.);; 2Key Laboratory of Potato Biology and Biotechnology (HZAU), Ministry of Agriculture and Rural Affairs, Wuhan 430070, China; 3Potato Engineering and Technology Research Center of Hubei Province, Huazhong Agricultural University, Wuhan 430070, China; 4College of Horticulture and Forestry Science, Huazhong Agricultural University, Wuhan 430070, China

**Keywords:** potato, bacterial wilt, *Ralstonia solanacearum*, wild resources, resistance

## Abstract

Potato is an important crop, used not only for food production but also for various industrial applications. With the introduction of the potato as a staple food strategy, the potato industry in China has grown rapidly. However, issues related to bacterial wilt, exacerbated by factors such as seed potato transportation and continuous cropping, have become increasingly severe in the primary potato cultivation regions of China, leading to significant economic losses. The extensive genetic diversity of *Ralstonia solanacearum* (*R. solanacearum*), which is the pathogen of bacterial wilt, has led to a lack of highly resistant potato genetic resources. There is a need to identify and cultivate potato varieties with enhanced resistance to reduce the adverse impact of this disease on the industry. We screened 55 accessions of nine different wild potato species against the bacterial wilt pathogen *R. solanacearum* PO2-1, which was isolated from native potato plants and belongs to phylotype II. Three accessions of two species (ACL24-2, PNT880-3, and PNT204-23) were identified with high resistance phenotypes to the tested strains. We found these accessions also showed high resistance to different phylotype strains. Among them, only PNT880-3 was capable of flowering and possessed viable pollen, and it was diploid. Consistent with the high resistance, decreased growth of *R. solanacearum* was detected in PNT880-3. All these findings in our study reveal that the wild potato PNT880-3 was a valuable resistance source to bacterial wilt with breeding potential.

## 1. Introduction

Potato is an essential crop with significant roles in food, vegetable, and industrial sectors, widely cultivated and consumed [1]. However, bacterial wilt, caused by *Ralstonia solanacearum*, poses a major threat to potato yields and quality [2]. The extensive genetic diversity within *R. solanacearum*, stemming from factors such as geographical distribution, host range, and pathogenicity, led to the application of the term ‘species complex’ for this organism [3]. This ‘*R. solanacearum* species complex’ (RSSC) consists of four phylotypes, each predominantly originating from different global regions. Phylotype I largely originates from Asia, phylotype II from America, phylotype III from Africa, and phylotype IV from Indonesia. Recent studies have redefined these phylotypes into separate species, including *R. solanacearum* (phylotype II), *R. pseudosolanacearum* (phylotypes I and III), and an array of *R. syzygii* (phylotype IV) [4]. In each phylotype, further classification based on the endoglucanase (*egl*) gene sequence, which is an intraspecific marker in the structure gene of *R. solanacearum*, delineates highly conserved strains within a specific gene range into sequevar. The strains utilized in this study are categorized using the phylotype-sequevar [3,5].

Potatoes belong to the genus *Solanum* in the section Petota (*Solanum* L. *Petota* Dumort) and originated in the Andes Mountains of South America. Potato genetic resources are abundant, although high inter-specific similarity exists. Based on morphological and molecular studies, the potato is divided into four cultivated and 107 wild species [6]. Potato genetic resources exhibit extensive genetic diversity, providing rich sources of genetic variation for breeding. Different approaches can be employed to transfer resistance alleles from these wild species into improved varieties, thus creating resistant cultivars [7].

Presently, the foremost resource for countering bacterial wilt that has been identified and widely used is *Solanum phureja* Juz. and Bukasov [8,9,10]. Researchers have innovated resistant genetic materials through techniques like hybrid breeding and protoplast fusion. *S. phureja*, recognized as the first cultivated germplasm resource with resistance to bacterial wilt, in conjunction with *S. phureja*–*S. tuberosum* hybrid varieties, displays diverse levels of resistance [11,12]. These revelations have led to the development of some cultivars that maintain stable resistance to bacterial wilt, such as MB-03 and MB9846-01 [13]. Dihaploid potato and *S. phureja* hybrids have demonstrated increased BW resistance [14]. A resistance allele was identified and an allele-specific molecular marker (Rbw6-1) for PBWR-6b was developed in the *S. phureja* [15,16].

*Solanum commersonii* Dunal is a wild potato species with a high tolerance to both biotic and abiotic stresses, and some genotypes of it have been proven to be resistant to bacterial wilt [17,18,19]. However, hybridization between *S. commersonii* and cultivated species is not easily achievable due to their interspecific incompatibility. Carputo used *S. phureja* as a bridge species to break the hybridization barrier between *S. commersonii* and *S. tuberosum* and successfully obtained hybrids. The hybrids showed good fertility and can be used for further breeding work [20]. Furthermore, some of the hybrids and introgressive lines from a backcross three progeny combined morphological traits from *S. tuberosum* and good resistance levels of *S. commersonii* [21]. Alternatively, through protoplast fusion, somatic hybrids were made that combine the *S. tuberosum* and *S. commersonii* genomes. Some of the somatic hybrid lines showed similar disease resistance levels to that of the resistant *S. commersonii* parent [22]. Researchers also demonstrated the resistance of *S. commersonii* to bacterial wilt and successfully transferred this resistance to cultivated potatoes through protoplast fusion [23]. In the wild species *S. chacoense*, some genotypes were subjected to somatic hybridization with cultivated species through protoplast fusion, resulting in some hybrid plants that displayed resistance to bacterial wilt [24].

Currently, the most efficient and direct approach for preventing bacterial wilt is the cultivation of varieties with resistance to bacterial wilt. Therefore, the search for potato germplasm resources with resistance to bacterial wilt is of utmost importance. In this study, we selected potato wild germplasm resources with potential resistance to bacterial wilt. For instance, *Solanum acaule* Bitter exhibits resistance to various potato diseases, including early blight, canker, and potato virus disease [7,25,26]. Furthermore, the resistance gene *Rx2* against Potato Virus X (PVX) was located on chromosome V [27]. *Solanum pinnatisectum* Dunal consistently exhibits strong resistance to various biotic stresses [28], particularly against late blight. Additionally, multiple disease-resistant genes have been located in this species, including *Rpi1* and *Rpi2* [29,30]. The wild species materials that we have screened belong to *S. pinnatisectum* (PNT880-3 and PNT204-23) and *S. acaule* (ACL24-2). Moreover, these materials exhibit strong resistance against various phylotype strains. Through phenotypic identification, we discovered that only PNT880-3 exhibited normal floral organs, and strong pollen vitality, and was identified as a diploid (2n). In addition, we found that this wild species PNT880-3 could limit the *R. solanacearum* multiplication during infection. In summary, this study identified a potential breeding resource that holds significant promise for its application in potato breeding for bacterial wilt resistance.

## 2. Results

### 2.1. The Traditional Identification of R. solanacearum Strains

In a previous study, three *Ralstonia solanacearum* strains were isolated from potato plants with bacterial wilt in the field, named Po2-1, Po2-2, and Po2-3 [31]. We utilized the specific flagellin gene (*fliC*) of the *R. solanacearum* for Polymerase Chain Reaction (PCR) testing, using the strains HA4-1 and UW551 as control samples. The results established that the isolated strains (Po2-1, Po2-2, and Po2-3) belong to the species of *R. solanacearum* (Figure 1a). To confirm the classification of the isolated strains, we employed multiple PCR using specific primers for phylotype. The results revealed the amplification of 372 bp and 280 bp bands, which are specific to phylotype IIB, matching the bands obtained from the phylotype IIB strains UW551 (Figure 1b).

We performed PCR amplification to obtain the respective 16S rDNA and *egl* segments from the isolated strains. Phylogenetic tree analysis was performed to compare three isolated strains with other strains that have been sequenced. As shown in Figure 1c,d, the results showed that the isolated strains clustered together with UW551 and IPO1609, which belong to sequevar 1. Therefore, we classified these isolated strains as phylotype IIB/sequevar 1.

### 2.2. Screening of Potato Resistance Resources

To screen potential potato germplasm with resistance to phylotype II strains, we selected 55 accessions of nine wild potato species (Table 1). We used the previously isolated *R. solanacearum* strain Po2-1 to evaluate the resistance levels of these wild potato species and used susceptible potato plants C9 as a control. The strain Po2-1 was classified as phylotype IIB/sequevar 1. The phylotype IIB/sequevar 1 was distributed throughout China in all four potato zones [31]. At 12 dpi, the potato C9701 exhibited symptoms of wilting and the whole plant showed a symptom of lodging. We used the disease index of each plant at 12 dpi to evaluate the resistance level to *R. solanacearum*. The results indicate a significant difference in resistance levels among the 55 wild potato species (Table 1). Notably, the genotypes ACL24-2, PNT880-3, and PNT204-23 belong to *S. acaule* and *S. pinnatisectum,* respectively, and showed strong resistance to strain Po2-1. We found that obvious resistance differences showed in these genotypes of the wild species *S. acaule* and *S. pinnatisectum* (Figure 2 and Figure 3).

To determine the resistance of the screened resistant genotypes to other *R. solanacearum*, they were inoculated with two other representative phylotype II strains (IPO160 and UW551) and two phylotype I strains (HA4-1 and PO46). The inoculation results showed that the potato species ACL24-2, PNT880-3, and PNT204-23 maintained a low disease index after inoculation with the test strains, indicating that the resistant materials we screened have good resistance to different *R. solanacearum* (Figure 2 and Figure 3).

### 2.3. PNT880-3 Displays the Breeding Value

Although the genotypes we screened were resistant to *R. solanacearum*, we observed that only the PNT880-3 was a plant capable of flowering. CMM5 is a diploid cold-resistant wild potato species *S. commersonii* [32], which we used as a control to detect the ploidy and pollen vigor of PNT880-3. We detected the ploidy of PNT880-3 by flow cytometry and a similar peak between PNT880-3 and CMM5 (Figure 4a). This result indicated that the PNT880-3 was a diploid wild potato species.

Next, we conducted a statistical analysis of the floral organs in PNT880-3. We mainly observed the shape of its petals and pollen grains. The flowers of PNT880-3 are white, and both the shape of the corolla and anthers are normal. The diameter of its pollen grains is slightly smaller than that of CMM5 (Figure 4b). After conducting pollen viability measurements, it was found that the average pollen viability of CMM5 is 84.3%, while PNT880-3 shows slightly higher average pollen viability at 85.9% (Figure 4c). These results indicate that PNT880-3 is a potential breeding resource with resistance to bacterial wilt.

### 2.4. PNT880-3 Limits R. solanacearum Multiplication in Root

Bacterial wilt resistance was associated with the capability of the plant to limit bacterial multiplication in the root. To detect the change in *R. solanacearum* population in the root after the inoculation of plants, we used the specific primers by the qRT-PCR method. We first drew a standard curve between the Ct value of the specific flagellin gene *fliC* and the population of the bacterial solution. It was found that the sCt value of *fliC* was negatively correlated with the bacterial population, and the coefficient of determination (R^2^) was 0.9991. Considering the efficiency of genome extraction kits and qRT-PCR, the standard curve is valid when the bacterial population is in the range of 10^4^–10^8^ CFU. We extracted the bacterial genome from the bacterial solution with known bacterial numbers and performed qRT-PCR to detect the Ct value of *fliC*. The results showed that which proved that the detection method of bacterial content was reliable (Figure 5a).

To quantify the bacterial population in the roots, we directly extract the bacterial genome from roots inoculated with the Po2-1 strain every two days. The results showed that within 12 days after inoculation, both the resistant material PNT880-3 and the susceptible material C9701 showed an increasing trend in the bacterial population within their roots. However, the bacterial population of *R. solanacearum* in PNT880-3 increased slowly and was always far lower than that in C9701 (Figure 5b,c). It indicated that the wild species PNT880-3 could limit the *R. solanacearum* multiplication during infection.

## 3. Discussion

Bacterial wilt is a disease that affects potato crops, so the discovery of wild species with resistance is of significant importance in agriculture. Here, we have screened and identified an accession PNT880-3 in wild species *S. pinnatisectum*, which exhibits extremely high resistance to bacterial wilt. There is currently no report about accession that confers resistance to bacterial wilt in *S. pinnatisectum*. This study holds promise for providing new resources and methods for breeding bacterial wilt-resistant potatoes.

The challenges in effectively preventing and controlling bacterial wilt stem from several factors: (1) the extensive genetic diversity within *R. solanacearum*, influenced by elements like geographic distribution, host range, and pathogenicity [3]; (2) although specific potato cultivars with partial resistance to bacterial wilt have been bred, this resistance is not absolute and can be gradually compromised by pathogen adaptation. [33]. Therefore, region-specific breeding of disease-resistant varieties is one of the main strategies to control the pathogen. In China, the *R. solanacearum* that poses a threat to potato production mainly belongs to phylotype IIB/sequevar 1 [31]. The phylotype IIB/sequevar 1 strains are highly pathogenic to potatoes and exhibit high toxicity under low-temperature conditions, which poses a severe threat to potato production in China [34,35]. The strain Po2-1 used in this study has also been identified as a phylotype IIB/sequevar 1 strain, which is a prevalent type in potato cultivation regions in China (Figure 1). Therefore, it is of great significance to screen for potato resources that are resistant to this type of strain.

The selected materials are potato wild germplasm resources with potential resistance to bacterial wilt [7]. While most wild species rapidly succumb to wilting upon inoculation, only three lines (ACL24-2, PNT880-3, and PNT204-23) from two wild potato species, *S. pinnatisectum* and *S. acaule* showed disease-resistant phenotypes (Figure 2). It is worth noting that ACL24-2 maintains strong resistance to the selected test strains. These three lines demonstrated excellent resistance against prevalent strains in China and showcased remarkable resistance across a wide range of bacterial wilt pathogens (Figure 3). This is the first case to identify resistance accessions from *S. pinnatisectum*, so they are considered high-quality germplasm resources for bacterial wilt resistance.

We discovered that the wild potato species PNT880-3 showed vigorous field growth, robust plants, and strong disease resistance (Figure 3 and Figure 4). After 20 days of inoculation, the survival rate of PNT880-3 remained at around 70% (Figure 3). Furthermore, the floral organs displayed a normal morphology, and the pollen grains were full and highly active (Figure 4). However, PNT880-3 is diploid (Figure 4a), while most commercial potato cultivars are autotetraploids [36]. Therefore, utilizing conventional hybrid breeding methods may lead to chromosome number and endosperm balance number (EBN) mismatch in embryos produced from interspecific potato crosses, potentially resulting in abnormal embryo development, which could lead to hybridization failure between them [37]. Crossing *S. pinnatisectum* directly with cultivars proved to be challenging and there have been limited reports on the utilization of *S. pinnatisectum*. To overcome EBN barriers, numerically unreduced (2n) gametes can be used to design breeding schemes aimed at equalizing the EBN of parents [38]. The 2n gametes are produced due to meiotic mutations during micro- and mega-sporogenesis and have been observed in various plant species across different taxa [39]. Hybrids were created by crossing wild potato species with the haploid form of tetraploid cultivated species [40] or diploid cultivated species such as *S. phureja* [41]. This approach allowed the offspring to inherit genetic diversity. Chromosome doubling can be induced through physical or chemical methods in polyploid induction, which can be used as a genetic bridge between individuals with different ploidy levels [42]. In cases of fertile pollen, conventional hybridization can be used to introduce desirable traits into cultivated potato varieties [43]. The resistance of diploid wild potato species to soft rot was transferred to cultivated varieties through hybridization methods [44].

In the resistance evaluation, ACL24-2 and PNT204-23 exhibited slightly higher resistance than PNT880-3, making them valuable resistance resources (Table 1). However, the absence of floral organs was observed in both of them. To utilize these materials, protoplast fusion technology can be employed separately. Protoplast fusion is an important tool for transferring genes for a desired quality and quantity of production [45]. The development and maturity of plant protoplast fusion technology provide a potential solution to issues related to the transfer of desirable traits between species due to reproductive isolation or hybridization incompatibility [46]. Somatic hybrids originating from the protoplast fusion between *S. tuberosum* and *S. chacoense* exhibited enhanced bacteria wilt resistance [47]. Remarkably, *S. tuberosum* and *S. stenotomum* hybrids have retained their parent’s resistance level even after five years of in vitro maintenance [48]. In summary, through a comprehensive approach that includes hybrid breeding, somatic hybridization technology, and other methods, it is possible to successfully incorporate the exceptional bacterial wilt resistance traits of ACL24-2, PNT204-23, and PNT880-3 into cultivated potato varieties, thus accomplishing the objective of resistance breeding.

In our study, we screened a broad array of wild potatoes against local isolates of bacterial wilt pathogen *R. solanacearum* and identified three materials from two species with a high degree of resistance. Additionally, these three materials also showed high resistance to different *R. solanacearum* strains, indicating that they are potential resistance resources. Among them, PNT880-3 has high-vigor pollen and can be used for cross-breeding. ACL24-2 and PNT204-23 were not capable of flowering and can be exploited by somatic hybridization. Some of these materials could be an important new source in potato breeding programs.

## 4. Materials and Methods

### 4.1. Plant and Bacterial Materials

*R. solanacearum* strains (PO2-1, PO2-2, PO2-3, UW551, IPO1609, HA4-1, PO46) are cultured on solid medium containing bactopeptone (10 g/L), casamino acids (1 g/L), and glucose (2.5 g/L) (BG medium) at 28 °C. Strains (Po2-1, Po2-2, and Po2-3) were isolated from potato plants with bacterial wilt in the field. Strains (UW551, IPO1609) are representative strains of phylotype IIB, and strains (HA4-1, PO46) are representative strains of phylotype I.

The potato wild species accessions were grown on Murashige–Skoog (MS) medium at 22 °C under a 16/8 h day/night photoperiod. The accessions of wild potato species were obtained from the International Potato Center (CIP). After 3 weeks, plants were sown in plastic boxes with a soil mix and cultivated at 18 to 25 °C in a glasshouse with a 16/8 h day/night cycle. C9 is a seeding clone of *S. chacoense* obtained from its natural pollination seeds [49]. It is very susceptible to *R. solanacearum* and has been used as a susceptible control [31].

### 4.2. Pathogenicity Assays

Potato plants were inoculated with 10 mL (OD_600_ = 0.1, corresponding to approximately 10^8^ cfu/mL) *R. solanacearum* suspension by soil drench [50]. One plant was placed in each plastic bowl, each replicate contained six plants. Symptoms were recorded every two days for three weeks after inoculation, using the following disease grade: 0 = plant without visible symptoms; 1 = 0 to 25% leaves wilted; 2 = 26% to 50% wilted; 3 = 51% to 75%; 4 = 76% to 100%. Concerning the resistance of the plants, six plants were inoculated and rated for disease grade, as described previously [31]. Disease index graphs were generated using GraphPad. Experiments were performed three times.

### 4.3. Bacteria DNA Extraction, and qRT-PCR

Total bacteria DNA was extracted from the inoculated roots using a TIANamp Bacteria DNA Kit (TIANGEN, DP302-02, Beijing, China). The qRT-PCR was performed on a LightCycler 480 II (Roche, Basel, Switzerland) with EvaGreen 2×qPCR MasterMix (ABM, http://www.abmgood.com (accessed on 5 December 2023)). In each case, PCR conditions were 95 °C for 15 min, followed by 40 cycles of 95 °C for 15 s, 60 °C for 30 s, and 72 °C for 30 s, as described previously [51]. The *R. solanacearum* flagellin (*fliC*) gene was used as a marker gene to normalize the bacterial population, and the standard curve between the Ct value of the *fliC* (F: GCTGTGGAATCCAACAACGG; R: GTTGGCGTTGGTTTCGATGT) and the population of the bacterial solution was calculated.

### 4.4. Pollen Vigor Analysis

The pollen viability of the PNT880-3 was estimated by staining pollen grains collected from house-grown glass plants. Pollen grains were placed on clean slides and stained with 20 μL acetocarmine (1%) in the dark, and then viable and non-viable pollen grains were counted and imaged by a fluorescence microscope (Zeiss Axioskop 40, Jena, Germany). After 3 min, the slide is examined under a microscope. Viable pollen appears red, circular, and well-developed, while sterile pollen remains uncolored, exhibits irregular shapes, and shriveled grains. Over six fields of view were randomly observed in each sample, as described previously [52].

### 4.5. Determination of the Ploidy

The ploidy level was determined using the CyStain^®^ UV Precise T Kit (Sysmex, 05-5003, Kōbe shi, Japan) by flow cytometry (BD FACSVerse, BD Biosciences, San Jose, CA, USA). About 0.5 cm^2^ of leaves from four-week-old plantlets were finely diced within a plastic Petri dish containing 60 μL of nucleus extraction solution for a duration of 30 to 60 s. Subsequently, the diced leaves were subjected to staining in 500 μL of Staining Buffer for 30 s. Afterward, the leaf extracts were filtered using a 50 μm CellTrics filter into a tube, and the DNA content of the isolated nuclei was assessed [53].

## Figures and Tables

**Figure 1 plants-13-00220-f001:**
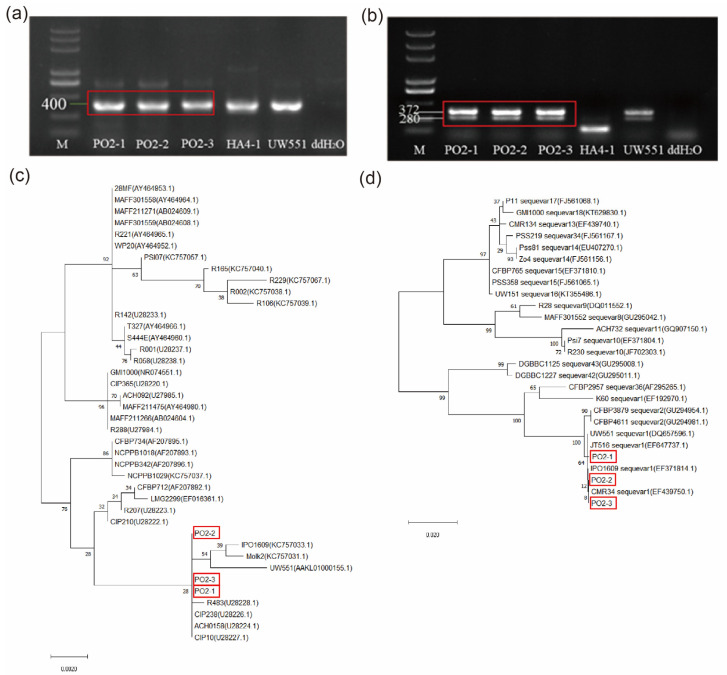
The isolated strains belong to Phylotype IIB sequence 1. (**a**) PCR results of the DNAs of the isolated strains with specific flagellin gene primers, the expected amplicon was 400 bp. (**b**) PCR results of the DNAs of the isolated strains with specific Species and Phylotypes gene primers, the expected Species amplicon was 280 bp, the expected Phylotype I amplicon was 144 bp, the expected Phylotype II amplicon was 372 bp, the expected Phylotype III amplicon was 213 bp, the expected Phylotype IV amplicon was 91 bp. (**c**) Phylogenetic analysis of isolates and other strains of *Ralstonia solanacearum* based on 16S rDNA gene sequences. (**d**) Phylogenesis of isolates and other strains of *Ralstonia solanacearum* based on egl gene sequences. Phylogenetic tree was constructed by the neighbor-joining method, bootstrap value was 1000. Red boxes indicate three isolated strains.

**Figure 2 plants-13-00220-f002:**
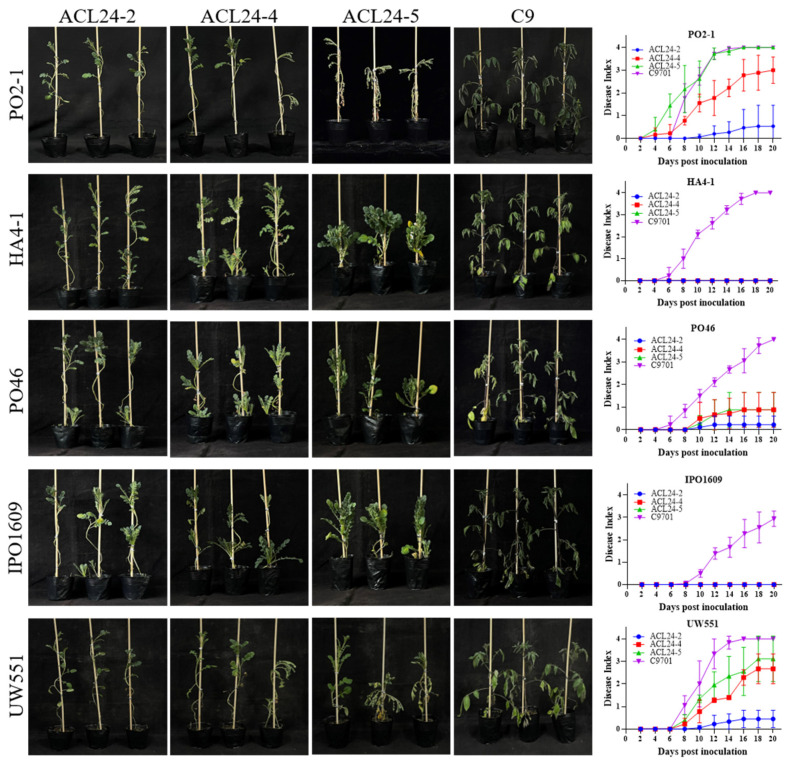
The accession ACL24-2 shows different resistance to *R. solanacearum* strains.The disease phenotype of wild potato species accession ACL24 inoculated with *Ralstonia solanacearum* PO2-1 and other four strains, photographed at 14 DPI, the disease index line chart of ACL24 series inoculated with strains on a single plant. The presented data are based on one representative experiment, comprising six plants per treatment. The error bars in the graph indicate standard errors. Three independent biological replicates were performed with similar results.

**Figure 3 plants-13-00220-f003:**
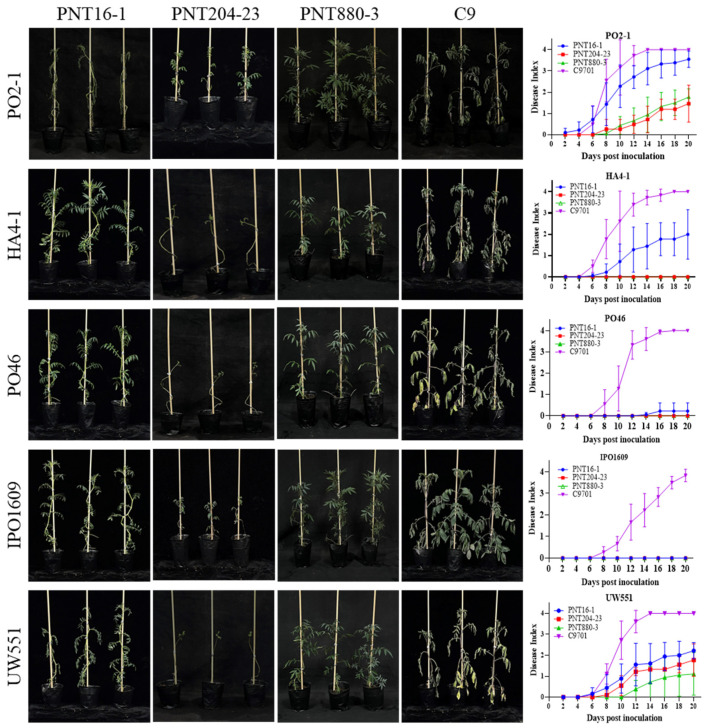
The accessions PNT204-23 and PNT880-3 show different resistance to *R. solanacearum* strains. The disease phenotype of wild potato species accessions PNT204-23 and PNT880-3 inoculated with *Ralstonia solanacearum* PO2-1 and other four strains, photographed at 14 DPI, the disease index line chart of potato inoculated with strains on a single plant. The presented data are based on one representative experiment, comprising six plants per treatment. The error bars in the graph indicate standard errors.

**Figure 4 plants-13-00220-f004:**
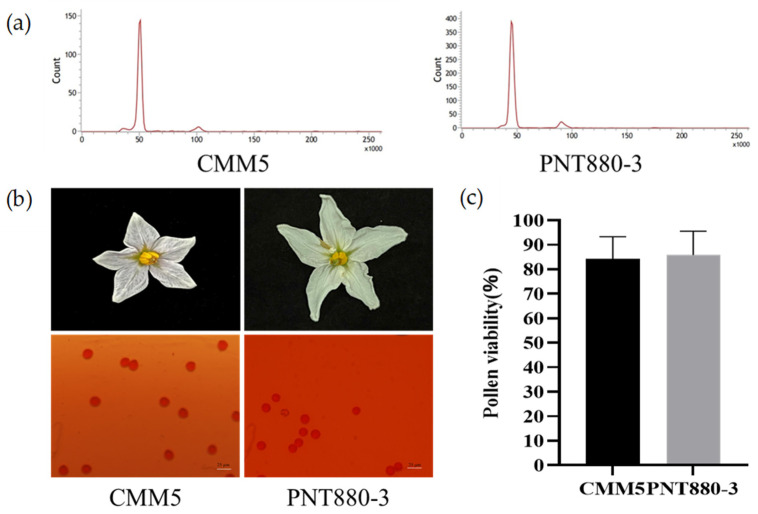
Plant PNT880-3 is diploid and pollen is viable. (**a**) PNT880-3 and CMM5 were analyzed using flow cytometry, showing peak values of 50, indicating that both are diploid. (**b**) Morphological observations of floral organs and pollen grains of PNT880-3 and CMM5 were conducted, with a scale bar of 25 μm. Pollen stainability was tested with 1% acetocarmine. (**c**) PNT880-3 displayed slightly higher pollen viability compared to CMM5. Error bars represent standard errors.

**Figure 5 plants-13-00220-f005:**
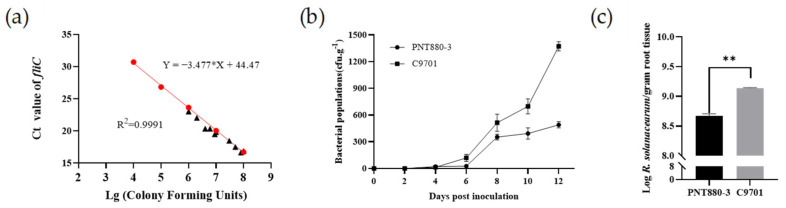
The populations of *R. solanacearum* in PNT880-3 were inhibited. (**a**) The Ct values of *fliC* expression in PO2-1 exhibited a negative correlation with the concentration of the bacterial solution, and the Ct values at different bacterial solution concentrations closely aligned with the standard curve. Red dots are the data for drawing a standard curve, and black triangles represent data for detecting the curve. (**b**) Quantitative analysis of the bacterial population of the bacterial wilt pathogen was conducted in resistant and susceptible materials 12 days after inoculation. (**c**) Significant differences in bacterial population were observed within the resistant and susceptible materials on the 12th day following plant inoculation. Error bars represent standard errors and two asterisks (**) = *p* < 0.01 by *t*-test.

**Table 1 plants-13-00220-t001:** Resistance rating of wild species materials.

Accessions ^a^	Species	Disease Index ^b^ (Mean ± SD)	Resistance Level ^c^
ACL24-2	*S. acaule*	0.53 ± 0.92	R
ACL24-4	*S. acaule*	1.78 ± 0.77	MR
ACL24-5	*S. acaule*	3.72 ± 0.25	S
ACL25-1	*S. acaule*	3.56 ± 0.38	S
ACL25-3	*S. acaule*	2.67 ± 0.67	MS
ACL25-4	*S. acaule*	3.72 ± 0.48	S
ACL27	*S. acaule*	3 ± 0.33	MS
ACP443-2	*S. acaule punae*	2.94 ± 0.35	MS
ALB26-1	*S. albicans*	2.28 ± 0.95	MS
ALB26-2	*S. albicans*	2.67 ± 1.26	MS
ALB26-3	*S. albicans*	2.22 ± 0.38	MS
ALB27-1	*S. albicans*	1.61 ± 1.42	MR
ALB27-2	*S. albicans*	1.67 ± 1.45	MR
ALB27-3	*S. albicans*	1.5 ± 1.69	MR
ALB28-1	*S. albicans*	3.72 ± 0.35	S
ALB28-2	*S. albicans*	3.11 ± 0.92	S
ALB28-3	*S. albicans*	3.17 ± 0.93	S
ALB462-2	*S. albicans*	3.17 ± 0.93	S
ALB464-3	*S. albicans*	1.56 ± 1.73	MR
RAP17-1	*S. raphanifolium*	3.47 ± 0.92	S
RAP17-2	*S. raphanifolium*	3.22 ± 1.07	S
RAP18-1	*S. raphanifolium*	3.78 ± 0.38	S
RAP19-3	*S. raphanifolium*	3.72 ± 0.48	S
RAP20-1	*S. raphanifolium*	2.78 ± 1.84	MS
RAP20-2	*S. raphanifolium*	2.8 ± 1.06	MS
STO80-1	*S. stoloniferum*	3.78 ± 0.38	S
STO80-3	*S. stoloniferum*	4 ± 0	S
STO80-5	*S. stoloniferum*	3.78 ± 0.38	S
STO80-6	*S. stoloniferum*	3.78 ± 0.38	S
BLB47-1	*S. bulbocastanum*	3.33 ± 1.15	S
BLB47-2	*S. bulbocastanum*	3.56 ± 0.77	S
BLB47-3	*S. bulbocastanum*	3.61 ± 0.67	S
PNT16-1	*S. pinnatisectum*	2.72 ± 0.54	MS
PNT204-23	*S. pinnatisectum*	0.49 ± 0.43	R
PNT880-3	*S. pinnatisectum*	0.67 ± 0.6	R
VRN90-1	*S. vernei*	94 ± 0	S
VRN90-2	*S. vernei*	4 ± 0	S
VRN90-5	*S. vernei*	3.72 ± 0.35	S
VRN904-6	*S. vernei*	3.06 ± 0.42	S
BET4-1	*S. berthaultii*	2.11 ± 1.51	MS
BET5-3	*S. berthaultii*	3.61 ± 0.35	S
BET5-5	*S. berthaultii*	4 ± 0	S
BET5-6	*S. berthaultii*	3.56 ± 0.38	S
BLV30-1	*S. boliviense*	2.11 ± 0.75	MS
BLV30-2	*S. boliviense*	2.89 ± 1.39	MS
BLV32-1	*S. boliviense*	4 ± 0	S
BLV32-2	*S. boliviense*	3.33 ± 1.15	S
BLV32-3	*S. boliviense*	2.67 ± 1.33	MS
BLV33-1	*S. boliviense*	3.28 ± 0.67	S
BLV33-3	*S. boliviense*	3.89 ± 0.1	S
BLV36-2	*S. boliviense*	3.67 ± 0.58	S
BLV36-3	*S. boliviense*	3.83 ± 0.29	S
BLV37-1	*S. boliviense*	3.2 ± 1.39	S
BLV37-2	*S. boliviense*	2.93 ± 1.1	MS
BLV37-3	*S. boliviense*	2.93 ± 1.85	MS

^a^ The accessions of wild potato species were obtained from the the International Potato Center (CIP). ^b^ Disease index (DI) was calculated as follows: DI = (n1 × 1 + n2 × 2 + n3 × 3 + n4 × 4)/(n0 + n1 + n2 + n3 + n4) × 4, where n0, n1, n2, n3, n4 represent the number of plants with symptoms of 0, 1, 2, 3 and 4 on the 21 days after inoculated, respectively. ^c^ The resistance level of potato to each strain was evaluated based on DI as follows: (i) 0 ≤ DI≤ 1 were considered resistant (R); (ii) DI values of 1 < DI ≤ 2 were considered resistant (MR); (iii) 2 < DI ≤ 3 as medium resistant (MS); (iv) 3 < DI ≤ 4 as susceptible (S).

## Data Availability

All data supporting the findings of this research are available within the paper.

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
