# Peer review of "Screening for Resistance Resources against Bacterial Wilt in Wild Potato"

_plants, 2024, doi:10.3390/plants13020220_

Round 1
Reviewer 1 Report (Previous Reviewer 1)
Comments and Suggestions for Authors
I check the manuscript again and found some week point I hope authors can reply on it
As I work with Ralstonia before in many papers I would like to ask authors 0.1 on spectrophotometer sure will not gave 108 CFU please could you check?
Also you inoculated the potato plants with 10 mL sure this will make damage for plant please check
in M&M you start to write the strain of the pathogen and you wrote this is pathogenic isolates how you know this is pathogenic and you still not make pathogenicity tests
What is the BG Agar ?
Please make sure that all scientific names is italic in the text
Author Response
I check the manuscript again and found some week point I hope authors can reply on it
Question 1: As I work with Ralstonia before in many papers I would like to ask authors 0.1 on spectrophotometer sure will not gave 108CFU please could you check?
Answer 1: We are grateful for the suggestion. We consulted two references on Ralstonia solanacearum inoculation. These two references were added in manuscript.
In Section 3.1 of the first reference, “Calculate the bacterial concentration, knowing that OD600nm = 1 corresponds to 109 CFU/mL”. (Morel, A., Peeters, N., Vailleau, F., Barberis, P., Jiang, G., Berthomé, R., & Guidot, A. (2018). Plant pathogenicity phenotyping of Ralstonia solanacearum strains. Host-Pathogen Interactions: Methods and Protocols, 223-239.)
In Section “Step-by-step method details” of the second reference, “For R. solanacearum, an OD600 of 0.1 corresponds to approximately 108 cfu/mL.” (Yu, G., Zhang, L., Wang, K., & Macho, A. P. (2023). Inoculation of Arabidopsis seedlings with Ralstonia solanacearum in sterile agar plates. STAR protocols, 4(3), 102474. DOI: 10.1016/j.xpro.2023.102474)
Question 2: Also you inoculated the potato plants with 10 mL sure this will make damage for plant please check.
Answer 2: We thank the reviewer for pointing out this issue. Potato plants in a pot were inoculated with 10 mL R. solanacearum suspension by soil drench. The bacterial solution is not injected directly into the plant but poured into the soil. This allows pathogenic bacteria to naturally invade the plant and facilitates more accurate detection of resistance.
Question 3: in M&M you start to write the strain of the pathogen and you wrote this is pathogenic isolates how you know this is pathogenic and you still not make pathogenicity tests.
Answer 3: Sorry, there's something wrong with our writing. We rewrote this sentence. “R. solanacearum strains (PO2-1, PO2-2, PO2-3, UW551, IPO1609, HA4-1, PO46) are cultured on BG medium containing bactopeptone (10 g/liter), casamino acids (1 g/liter), and glucose (2.5 g/liter) at 28°C.”
Question 4: What is the BG Agar?
Answer 4: We apologize for our mistake. We wrote it wrong, it is BG medium.
Question 5: Please make sure that all scientific names is italic in the text.
Answer 5: We are very sorry for our incorrect writing. We have made corrections.

Reviewer 2 Report (New Reviewer)
Comments and Suggestions for Authors
The bacterial wilt is a major challenge in potato cultivation and so far, no potential resistant cultivar is available worldwide. This study suggests some wild accessions having potential resistance to this dreadful disease. It is valuable work and studies but I have some queries and suggestions
1. Material and method section 4.1 please describe the source of these strains. improve grammar.
2. Section 4.2: this is not the way of writing methodology in a research paper.
3. Statistical analysis details and designs of experiment is missing.
4. How this inoculation method was standardized? Was it previously published somewhere else? It is very difficult on artificially inoculate the plants and observe the symptoms in case of this disease.
5. Figure 2 should be bifurcated and the plant picture should be enlarged for better visibility of symptoms. Likewise, the graphs should be enlarged and kept as separate figure.
Comments on the Quality of English LanguageCorrect the grammatical errors.
Author Response
The bacterial wilt is a major challenge in potato cultivation and so far, no potential resistant cultivar is available worldwide. This study suggests some wild accessions having potential resistance to this dreadful disease. It is valuable work and studies but I have some queries and suggestions
- Material and method section 4.1 please describe the source of these strains. improve grammar.
Answer 1: We thank the reviewers for pointing out the lack of information. The information for these strains has been added. “Strains (Po2-1, Po2-2, and Po2-3) were isolated from potato plants with bacterial wilt in the field. Strains (UW551, IPO1609) are representative strains of phylotype IIB, and strains (HA4-1, PO46) representative strains of phylotype I.”
- Section 4.2: this is not the way of writing methodology in a research paper.
Answer 2: Thanks for your nice suggestions. We agree with the comment and rewrote the sentence in the revised manuscript. “Potato plants were inoculated with 10 mL (OD600 = 0.1, corresponds to approximately 108 cfu/mL) R. solanacearum suspension by soil drench. And then their symptoms were recorded every 2 days for continuous 3 weeks after inoculation, using the following disease grade: 0 = plant without visible symptoms; 1 = 0 to 25% leaves wilted; 2 = 26% to 50% wilted; 3 = 51% to 75%; 4 = 76% to 100%. Concerning the resistance of the transgenic plants, 6 plants were used for the investigation of the disease index (DI), as described previously. Generate a disease index graph using GraphPad. Experiments were performed three times.”
- Statistical analysis details and designs of experiment is missing.
Answer 3:. Thank you for pointing out this problem in manuscript. The information for these strains has been added to figure legends and “Materials and Methods”.
- How this inoculation method was standardized? Was it previously published somewhere else? It is very difficult on artificially inoculate the plants and observe the symptoms in case of this disease.
Answer 4: We thank the reviewer for this important suggestion. This inoculation method requires controlling the environment and the growth status of the plants. Therefore, we will select plants with consistent growth status for resistance testing, and the enviromment were controlled by the a climate chamber. We have published several articles applying this method.
Wang, L.; Wang, B.S.; Zhao, G.Z.; Cai, X.K.; Jabaji, S.; Seguin, P.; Chen, H.L. (2017). Genetic and Pathogenic Diversity of Ralstonia solanacearum Causing Potato Brown Rot in China. Am J Potato Res
Wang, B., Wang, Y., He, W., Huang, M., Yu, L., Cheng, D., ... Song, B., & Chen, H. (2023). StMLP1, as a Kunitz trypsin inhibitor, enhances potato resistance and specifically expresses in vascular bundles during Ralstonia solanacearum infection. The Plant Journal.
Huang, M., Tan, X., Song, B., Wang, Y., Cheng, D., Wang, B., & Chen, H. (2023). Comparative genomic analysis of Ralstonia solanacearum reveals candidate avirulence effectors in HA4-1 triggering wild potato immunity. Frontiers in Plant Science.
- Figure 2 should be bifurcated and the plant picture should be enlarged for better visibility of symptoms. Likewise, the graphs should be enlarged and kept as separate figure.
Answer 5: We are grateful for the suggestion. There is a lot of data in this figure, so we will choose high-definition pictures so that readers can zoom in and observe.

Reviewer 3 Report (New Reviewer)
Comments and Suggestions for Authors
Lines 131-134: why using this isolate only and how representative is this isolate of the R. solanacearum pop in China.
Line 171: under which conditions? Which other conditions. Were used or have been used in past to address flowering issues in potato wild species? Were the conditions too limiting?
Lines 294-297: more information about origin of isolates and their virulence is needed. See comments on lines. 131-134 too.
Lines 308: It is not clear how many plants per replicate.
Comments on the Quality of English Language
Extensive grammar revisions are needed.
Author Response
Question 1: Lines 131-134: why using this isolate only and how representative is this isolate of the R. solanacearum pop in China.
Answer 1: Thanks for the reviewer’s suggestion. The three strains we isolated originated from the same area, and the identification results are similar. The strain Po2-1 was classified as phylotype IIB/sequevar 1. The phylotype IIB/sequevar 1 represented 90% of all strains isolated and it was distributed throughout China in all four potato zones (Wang et al, 2017).
Wang, L.; Wang, B.S.; Zhao, G.Z.; Cai, X.K.; Jabaji, S.; Seguin, P.; Chen, H.L. Genetic and Pathogenic Diversity of Ralstonia solanacearum Causing Potato Brown Rot in China. Am J Potato Res 2017, 94, 403-416, doi:10.1007/s12230-017-9576-2.
Question 2: Line 171: under which conditions? Which other conditions. Were used or have been used in past to address flowering issues in potato wild species? Were the conditions too limiting?
Answer 2: We thank the reviewer for this suggestion. Potato plants were sown in plastic boxes with a soil mix and cultivated at 18 to 25°C in a glasshouse with a 16/8 h day/night cycle. This is a normal growing condition. Our purpose is to conduct a preliminary identification of the screened materials. PNT880-3 can flower under normal conditions, only proving that it has certain advantages. We will also use other methods to apply other resistant materials, this part we have added in Discussion.
Question 3: Lines 294-297: more information about origin of isolates and their virulence is needed. See comments on lines. 131-134 too.
Answer 3: We thank the reviewers for pointing out the lack of information. The information for these strains has been added. “Strains (Po2-1, Po2-2, and Po2-3) were isolated from potato plants with bacterial wilt in the field. Strains (UW551, IPO1609) are representative strains of phylotype IIB, and strains (HA4-1, PO46) representative strains of phylotype I.”
Question 4: Lines 308: It is not clear how many plants per replicate.
Answer 4: Sorry for not writing it clearly, we have corrected it. “one plant in each plastic bowl, each replicate containing six plants.”

Round 2
Reviewer 1 Report (Previous Reviewer 1)
Comments and Suggestions for Authors
Ok
Author Response
Thank you for your comments that help us improve the quality of the manuscript.
Reviewer 2 Report (New Reviewer)
Comments and Suggestions for Authors
All comments were nicely addressed.
Author Response
Thank you for your comments that help us improve the quality of the manuscript.
This manuscript is a resubmission of an earlier submission. The following is a list of the peer review reports and author responses from that submission.
Round 1
Reviewer 1 Report
Comments and Suggestions for Authors
I think authors need to rewrite the paper again
Sometime authors write the title of Fig in the top and sometime in the bottom the same with the table materials and methods missing the most important details
Keyword should be different than in title
Make sure that all scientific names in the References list are italics.
Please add the DOI for ALL the References
All tables must be self-explanatory
Author Response
I think authors need to rewrite the paper again
Answer: We apologize for the confusion in the manuscript. We tried our best to improve and made some changes to the manuscript.
Sometimes authors write the title of Fig in the top and sometimes in the bottom the same with the table materials and methods missing the most important details
Answer: Thank you for pointing out this problem in the manuscript. We tried our best to improve the manuscript and made some changes in the manuscript.
Keyword should be different than in title
Answer: Thanks to the reviewer for pointing this issue out. We have added the Keywords in the Abstract
Make sure that all scientific names in the References list are italics.
Please add the DOI for ALL the References
Answer: We apologize for the problems in the References. We tried our best to improve the references in the manuscript.
All tables must be self-explanatory
Answer: Thanks for the reviewer’s suggestion. We added some legends in Table 1.

Reviewer 2 Report
Comments and Suggestions for Authors
Manuscript ID
plants-2635763
REVIEW of the MS by Wenfeng He et al. “Screening for resistance resources against bacterial wilt in wild potato”.
Potato is one of the most important agricultural crops both in China and almost all over the world. The topic of the paper is highly important in connection with the one of the most destructive potato diseases - bacterial wilt that caused by Ralstonia solanacearum. Therefore, the search for new sources of resistance to bacterial wilt is relevant for future potato breeding programs. This paper provides a good example of the value of wild potato relatives for potato breeding. In this study, W.He et al. screened the wide subset of wild potato species accessions for bacterial wilt resistance. They identified three accessions with high resistance to R. solanacearum and discussed the possibilities of transferring this resistance trait into cultivated potato through interspecific hybridization and introgression. The important part of this research is the assessment of genetic variability of this bacterial pathogene in China using both traditional and molecular methods to identify R. solanacearum strains.
Unfortunately, authors did not discuss these really important results (see Discussion part).
It is not clear why the authors made a ploidy level determination for S. commersonii and S. pinnatisectum accessions - it is well-known that these wild species are diploids.
From our point of view ‘Discussion’ needs to be rewritten – (1) the main emphasis should be placed on the results of phylogenetic & phenotypic analyses of R. solanacearum and comparing them with literature data; (2) Please make the part on interspecific hybridization more concise, since it relates not to the results of your paper, but to plans for the future.
Also, there are many spelling and typos errors, inaccuracies in terms that need to be corrected. Some specific points:
Line 23. Author citation for species name Ralstonia solanacearum is lost.
Line 30, Line 281. ‘Fertile pollen’ or ‘ viable pollen’ instead of “active pollen’ Line 35. Keywords are absent.
Line 50. Taxa names need to use Author citation - Solanum L., Petota Dumort. The same for
Line 57 (S. phureja, S. commersonii), Line 77 (S. acaule), Line 78 (S. pinnatisectum).
Line 50. The section Petota instead of the group Petota.
Line 54. ‘transfer’ instead of ‘integrate’
Line 64. ‘interspecific incompatibility’ instead of ‘hybridization incompatibility’.
Line 66. It is not clear what the authors mean when they talk about ‘hybrid varieties’. The articles they cited here refer to hybrids and introgressive lines from backcross progenies.
Line 69. It is not clear what the authors mean here – ‘resistance characteristics akin to those of S. commersonii’.
Line 69. Species name need to use italic font: S. commersonii’.
Line 78. Why ‘also’ ?
Lines 78-80. The authors mentioned genes that confer potato resistance to viruses and late blight, but did not provide literature data about QTLs that confer resistance to bacterial wilt.
Lines 102 & 103. ‘Phylogenetic analysis’ instead of ‘Phylogenesis’ should be given here.
Line 103 ‘Phylogenetic tree’ instead of ‘The evolutionary tree’ should be given here.
Line 111. More information needs to be provided on the potato plant C9 used as a susceptible control – here or in the part ‘Research Material’.
-Page 6 - Table 1, Columne 2. I don’t see any ‘hybrid crosses‘ in columne 2.
Lines 164 & Line 252. According to the data presented in the paper, the authors stated the ability of plants to flower, but did not really investigate the reproductive stage – peculiarities of developing of individual reproductive organs (sepals, petals, anthers, pestles, ….). So, it’s better indicate in the text “flowering plant” or ‘’plants capable of flowering” instead of - ‘we observed the presence of floral organs’ - Lines 164. The same – use ‘plants were not capable of flowering’ instead of ‘they ‘do not have complete floral organs’ - Line 252.
Some potato genotypes derived from in vitro culture often do not form flowers during the first year of their growing ex vitro. However after one year’ adaptation to ex vitro conditions they may have good flowering ability. The question is: observations of plant development were carried out for one season or over 1 – 2 years?
Line 174. Figures 4A, 4B shows too little information, because it is well-known that S. commersonii and S. pinnatisectum are diploids. So placing it in the supplement is more appropriate. The same for Figures 4C, 4D - there is enough information in the text about the high fertility level of the studied accessions. The information about high pollen fertility is already mentioned in the text.
Line 259. Insert please ‘accessions” - The potato wild species ACCESSIONS were grown on Murashige-Skoog (MS) medium”.
Line 259. it is necessary to indicate from which genebank the accessions of wild potato species were obtained.
Line 282. Sterile pollen instead of ‘inactive pollen’.
To conclude, revision should be made - I can recommend to revise this paper.

Moderate editing of English language required for all text
Author Response
Manuscript ID
plants-2635763
REVIEW of the MS by Wenfeng He et al. “Screening for resistance resources against bacterial wilt in wild potato”.
Potato is one of the most important agricultural crops both in China and almost all over the world. The topic of the paper is highly important in connection with the one of the most destructive potato diseases - bacterial wilt that caused by Ralstonia solanacearum.
Therefore, the search for new sources of resistance to bacterial wilt is relevant for future potato breeding programs. This paper provides a good example of the value of wild potato relatives for potato breeding. In this study, W.He et al. screened the wide subset of wild potato species accessions for bacterial wilt resistance. They identified three accessions with high resistance to R. solanacearum and discussed the possibilities of transferring this resistance trait into cultivated potato through interspecific hybridization and introgression. The important part of this research is the assessment of genetic variability of this bacterial pathogene in China using both traditional and molecular methods to identify R. solanacearum strains.
Unfortunately, authors did not discuss these really important results (see Discussion part).
It is not clear why the authors made a ploidy level determination for S. commersonii and S. pinnatisectum accessions - it is well-known that these wild species are diploids.
Answer: Thanks for your nice suggestions. Although this information about S. pinnatisectum is well-known, in order to further utilize PNT880-3, these detections were required. It is to confirm the ploidy of PNT880-3 through ploidy detection, the S. commersonii accession is a control.
From our point of view ‘Discussion’ needs to be rewritten – (1) the main emphasis should be placed on the results of phylogenetic & phenotypic analyses of R. solanacearum and comparing them with literature data; (2) Please make the part on interspecific hybridization more concise, since it relates not to the results of your paper, but to plans for the future.
Answer: We apologize for the confusion in the Discussion. We tried our best to improve and made some changes in the Discussion.
Also, there are many spelling and typos errors, inaccuracies in terms that need to be corrected.
Some specific points:
Line 23. Author citation for species name Ralstonia solanacearum is lost.
Answer: Thanks for the suggestion. Ralstonia solanacearum was first described by Erwin F. Smith in 1896 and was first named Bacillus solanacearum. However, when publishing articles about Ralstonia solanacearum, it is generally not necessary to indicate the Author citation.
Line 30, Line 281. ‘Fertile pollen’ or ‘ viable pollen’ instead of “active pollen’ .
Line 35. Keywords are absent.
Line 50. Taxa names need to use Author citation - Solanum L., Petota Dumort. The same for
Line 57 (S. phureja, S. commersonii), Line 77 (S. acaule), Line 78 (S. pinnatisectum).
Line 50. The section Petota instead of the group Petota.
Line 54. ‘transfer’ instead of ‘integrate’
Line 64. ‘interspecific incompatibility’ instead of ‘hybridization incompatibility’.
Answer: We have corrected the problems as you suggested and added the Keywords in manuscript.
- Line 66. It is not clear what the authors mean when they talk about ‘hybrid varieties’. The articles they cited here refer to hybrids and introgressive lines from backcross progenies.
Answer: We apologize for the confusion and rewrote this sentence. “Carputo used S. phureja as a bridge species to break the hybridization barrier between S. commersonii and S. tuberosum and successfully obtained hybrids. The hybrids showed good fertility and can be used for further breeding work. Furthermor, some of the hybrids and introgressive lines from a backcross 3 progeny combined morphological traits from S. tuberosum and good resistance levels of S. commersonii.”
- Line 69. It is not clear what the authors mean here – ‘resistance characteristics akin to those of S. commersonii’.
Answer: We apologize for the confusion and rewrote this sentence. “The some of the somatic hybrid lines showed similar disease resistance levels to that of the resistant S. commersonii parent.”
Line 69. Species name need to use italic font: S. commersonii’.
Line 78. Why ‘also’ ?
Answer: We apologize for our mistake. We corrected the italics and removed “also”.
- Lines 78-80. The authors mentioned genes that confer potato resistance to viruses and late blight, but did not provide literature data about QTLs that confer resistance to bacterial wilt.
Answer: Because the materials we screened belong to Solanum acaule and Solanum pinnatisectum, there is currently no literature data about QTLs that confer resistance to bacterial wilt in these two wild potato species. We showed the research progress of other diseases.
Lines 102 & 103. ‘Phylogenetic analysis’ instead of ‘Phylogenesis’ should be given here.
Line 103 ‘Phylogenetic tree’ instead of ‘The evolutionary tree’ should be given here.
Answer: We have corrected the problems as you suggested.
- Line 111. More information needs to be provided on the potato plant C9 used as a susceptible control – here or in the part ‘Research Material’.
Answer: We have added the information in ‘Materials and Methods’. “C9701 is a seeding-clone of S. chacoense obtained from its natural pollination seeds. It is very susceptible to Ralstonia solanacearum and has been used as a susceptible control.”
Page 6 - Table 1, Columne 2. I don’t see any ‘hybrid crosses‘ in columne 2.
Answer: We apologize for our mistake and removed the ‘hybrid crosses’.
- Lines 164 & Line 252. According to the data presented in the paper, the authors stated the ability of plants to flower, but did not really investigate the reproductive stage – peculiarities of developing of individual reproductive organs (sepals, petals, anthers, pestles, ….). So, it’s better indicate in the text “flowering plant” or ‘’plants capable of flowering” instead of - ‘we observed the presence of floral organs’ - Lines 164. The same – use ‘plants were not capable of flowering’ instead of ‘they ‘do not have complete floral organs’ - Line 252.
Answer: We are grateful for the suggestion and have corrected in manuscript.
- Some potato genotypes derived from in vitro culture often do not form flowers during the first year of their growing ex vitro. However after one year’ adaptationto ex vitro conditions they may have good flowering ability. The question is: observations of plant development were carried out for one season or over 1 – 2 years?
Answer: We appreciate the comments from the reviewer. Considering further utilization of these three accessions, our observations of plant development over two years.
- Line 174. Figures 4A, 4B shows too little information, because it is well-known that S. commersonii and S. pinnatisectum are diploids. So placing it in the supplement is more appropriate. The same for Figures 4C, 4D - there is enough information in the text about the high fertility level of the studied accessions. The information about high pollen fertility is already mentioned in the text.
Answer: Thanks for your nice suggestions, and we agree with the comment. Although this information about S. pinnatisectum is well-known, in order to further utilize PNT880-3, these detections were required.
Line 259. Insert please ‘accessions” - The potato wild species ACCESSIONS were grown on Murashige-Skoog (MS) medium”.
Answer: Thanks for the reviewer’s suggestion, we added the word in this sentence. Q. Line 259. it is necessary to indicate from which genebank the accessions of wild potato species were obtained.
Answer: We thank the reviewer for this suggestion. This sentence was added in the Materials and Methods. “The accessions of wild potato species were obtained from the International Potato Center (CIP).”
Line 282. Sterile pollen instead of ‘inactive pollen’.
Answer: We are grateful for the suggestion and have corrected in manuscript.
To conclude, revision should be made - I can recommend to revise this paper.

Reviewer 3 Report
Comments and Suggestions for Authors
Please see attached file

Author Response
Manuscript ID: plants-2635763
Title: Screening for resistance resources against bacterial wilt in wild 2 potato
This manuscript describes the screening of wild species having resistance to bacterial wilt. Bacterial wilt caused by R. solanacearum is one of the major threats to potato production especially in warm region. The most effective measures to control the disease is the cultivation of resistant cultivars, so identification of resistance resources and use in breeding program are important. The resistance materials identified in this study will contribute breeding of new resistant cultivars. However, screening of resistant germplasm from wild species has been widely conducted. This manuscript needs to be revised to clearly show what is new compared to previous work. I have commented on some points that need consideration or correction.
- As pointed above, screening of resistant sources from cultivated and wild species have been widely carried out and several species have been identified as resistant. The manuscript must be revised to indicate novelty of this research. As far as I know, this is first case to identify resistance clone from pinnatisectum. Please review the previous studies and clearly indicate what is new about this study compared to those studies.
Answer: This comment is all valuable and very helpful for revising and improving our paper, as well as the important guiding significance to our researches. Related contents have been added to the part ‘Discussion’. “Bacterial wilt is a disease that affects potato crops, so the discovery of wild species with resistance is of significant importance in agriculture. Here, we have screened and identified an accession PNT880-3 in wild specice S. pinnatisectum, which exhibits extremely high resistance to bacterial wilt. There is currently no report about accessions that confer resistance to bacterial wilt in S. pinnatisectum. This study holds promise for providing new resources and methods for breeding bacterial wilt-resistant potato.”
- There are too few references cited. For example, there are many more studies on resistance from phureja, and there are varieties that have introduced that resistance, so they should be cited. Here are some examples : Fock et al. 2000, French et al. 1998, French and De Lindo 1982, Habe et al., 2023, Mori et al. 2012 and Sakamoto et al. 2017. In other parts of the manuscript, there are also too few references, so please include references as appropriate.
Answer: We have added many appropriate reference as the reviewer's suggestion.
- Figure 2 and 3. The figure is improperly constructed and difficult to understand. I think panels A and B are doing the same experiment, just with different inoculum. Why are they divided into A and B? Also, I assume that you inoculated the species of bacteria shown in the graph on the right side of each picture, but please indicate this clearly.
Answer: We are grateful for the suggestion. We have readjusted the images and also specified the species of inoculated strains.
- Figure 2 and 3. Some of the pictures do not appear to match the data in the graphs. For example, the picture of PNT204-23 inoculated with HA4-1 and PO46 appears to have a higher DI than that of PO2-1.
Answer: Thanks for the suggestion. After inoculation, PNT204-23 exhibited a continuous shedding of its lower leaves, but this leaf loss was not indicative of symptoms related to bacterial wilt. Because, after 20 days post-inoculation, the upper leaves of the plant remained viable, and its stem continued to grow. We primarily assessed the Disease Index (DI) based on the wilting degree of the plant's own leaves, and there were no signs of wilting in the upper leaves after PNT204-23 inoculated with HA4-1 and PO46. Overall, following inoculation with HA4-1 and PO46, PNT204-23 had a lower DI compared to the case where PO2-1 was inoculated, as some of the PNT204-23 plants wilted after being inoculated with PO2-1, while there were no plant deaths following inoculation with HA4-1 and PO46.
Minor point
- Italics are not used properly. Species names should be italicized. There seems to be a lot of inappropriate writing, especially in cited references.
Answer: We apologize for the inappropriate writing in the original manuscript. We tried our best to improve the manuscript and made some changes in the manuscript.
- Table1 Lines are too narrow and text cannot be read properly. Please widen the width.
Figure 4C There are Chinese characters.
Line 244 BW should be bacterial wilt.
Answer: We apologize for our mistakes and have corrected the problems.

Round 2
Reviewer 1 Report
Comments and Suggestions for Authors
All Figure need to improve
still some of my comments not cover especially for the title of the Fig
the title of the table should be in the top of the Table